# An Ensemble Learning Solution for Predictive Maintenance of Wind Turbines Main Bearing

**DOI:** 10.3390/s21041512

**Published:** 2021-02-22

**Authors:** Mattia Beretta, Anatole Julian, Jose Sepulveda, Jordi Cusidó, Olga Porro

**Affiliations:** 1Unitat Transversal de Gestió de l’Àmbit de Camins UTGAC, Universitat Politécnica de Catalunya, 08034 Barcelona, Spain; mattia.beretta@smartive.eu; 2SMARTIVE S.L., 08204 Sabadell, Spain; anatole.julian@smartive.eu (A.J.); jose.sepulveda@smartive.eu (J.S.); olga.porro@smartive.eu (O.P.); 3Enginyeria de Projectes i de la Construcció EPC, Universitat Politécnica de Catalunya, 08028 Barcelona, Spain; 4Facultat de Matemàtiques i Estadística, Universitat Politécnica de Catalunya, 08028 Barcelona, Spain

**Keywords:** main bearing, wind turbine, failures, predictive maintenance, ensemble learning, unsupervised, interpretable, scalable, SCADA

## Abstract

A novel and innovative solution addressing wind turbines’ main bearing failure predictions using SCADA data is presented. This methodology enables to cut setup times and has more flexible requirements when compared to the current predictive algorithms. The proposed solution is entirely unsupervised as it does not require the labeling of data through work orders logs. Results of interpretable algorithms, which are tailored to capture specific aspects of main bearing failures, are merged into a combined health status indicator making use of Ensemble Learning principles. Based on multiple specialized indicators, the interpretability of the results is greater compared to black-box solutions that try to address the problem with a single complex algorithm. The proposed methodology has been tested on a dataset covering more than two year of operations from two onshore wind farms, counting a total of 84 turbines. All four main bearing failures are anticipated at least one month of time in advance. Combining individual indicators into a composed one proved effective with regard to all the tracked metrics. Accuracy of 95.1%, precision of 24.5% and F1 score of 38.5% are obtained averaging the values across the two windfarms. The encouraging results, the unsupervised nature and the flexibility and scalability of the proposed solution are appealing, making it particularly attractive for any online monitoring system used on single wind farms as well as entire wind turbine fleets.

## 1. Introduction

The future is bright for wind energy. New turbines are being installed, technologies are improving and costs are decreasing. IRENA estimates a tumultuous growth for the industry, expecting a global installed capacity of 1000 GW by 2050, and new installations rate of 200 GW/yr, including replacement of old turbines [1]. By the end of 2019 Europe alone boasted 205 GW of installed wind power capacity [2].

A number of challenges have to be faced in order to reach such ambitious goals, reducing costs of operation and maintenance (O&M) is paramount. In large wind farms O&M costs can account up to 30% of the total cost of energy, the influence of physical maintenance is estimated around 20% of the levelized cost of electricity (LCOE) [3]. Turbines are often situated in remote locations, the components are bulky and difficult to transport, logistics costs are significant. The growth of offshore installation, which accounted for 22 GW of power capacity in Europe in 2019 [2], exacerbates the problem as logistics becomes even more challenging.

Of all the components in a turbine: the main bearing, which provides support to the ma axis connecting blades and gearbox, is one of the most problematic in terms of maintenance and logistics. Failure rates reaching 30% for single main-bearing and of 15% for double main-bearing turbines, over 20 year lifetime are reported by Hart et al. [4]. Replacing a main bearing is no trivial task, unlike other systems that can be repaired in on-tower interventions, a crane is needed and the faulty turbine has to be put out of production for a long period of time.

Most turbines are equipped with a Supervisory Control and Data Acquisition (SCADA) system. This is a network of sensors monitoring various physical quantities: such as temperature, speed and pressure of the principal components of a turbine. International Standards, such as IEC 61400-25 simplify the representation of turbines and guarantee the uniformity of information exchange and control design [5]. While initially designed for control purposes only, SCADA data have also predictive capabilities and it has been used widely in the literature [6,7].

This articles presents a solution built on SCADA data to address main bearing failures, predicting the occurrence of future faults, and thus, helping wind farm operators to improve maintenance and reduce costs related to unexpected failures. Predictions from a set of understandable indicators, designed to capture different characteristics of the signal, are combined into a composed health status indicator. Data from two onshore wind farms, for a total of 84 monitored turbines, is used to evaluate the performances of this solution.

The main contributions of this research can be summarized in three key-points:Present an unsupervised system, requiring minimum setup and limited prerequisites, capable to monitor entire wind farms.Provide interpretable and understandable predictions, in contrast to black-box solutions.Implement an Ensemble Learning strategy that produces reliable predictions from a set of understandable indicators, improving their individual performances.

### 1.1. Main-Bearing Failure Discussion

The rolling elements of wind turbines’ main-bearing are subjected to severe working conditions, far different from the typical stress that are known in other industrial applications such as power plants. Windspeed, turbulence index and in general variations of the wind field conditions have a significant effect on main bearing deterioration [4].

The principal damage and wear mechanisms are reported by Hart et al. [8], defects in the assembly, design and manufacturing of main bearings lead to premature wear of the main bearing. Phenomenons such as micro-pitting, spalling, smearing etc. can be observed [8]. Progressive wear of material leads to sub-optimal operating conditions, higher localized loads where defects arise and in general overheating of the main bearing.

An incipient main-bearing failure is expected to be preceded by anomalous vibrations and increases in temperature of the component. In this study, vibration measurements are not available, thus the attention is given to anomalous patterns in temperature readings. Moreover, temperature signals are easy to interpret and they are part of the typical recordings of a SCADA system, unlike vibration signals that rarely are available. The use of temperature data thus make this solution applicable for a wider range of wind-farms.

Different authors have successfully studied temperature behaviors to predict failures in various turbines’ components. Guo et al. devised a monitoring strategy for turbines’ generators based on tracking of the generator temperature via change detection of a memory matrix of the component behavior [9]. Qiu et al. presented a thermophysics approach to assess drive train conditions from which various diagnostic rules are defined [10]. Tonks and Wang showed experimentally that monitoring temperature can reveal misalignments and problems of shaft couplings, as these defects increase friction therefore temperature of the component [11]. Cambron et al. developed a method to monitor main bearing condition comparing the measured and expected temperture of the component, predictions were obtained using a physical model of the bearing [12]. Sun et al. describe an anomaly identification method using mainly temperature readings and other standard SCADA signals to monitor the behavior of the major components [13].

One of the main bearing failure event is presented. Figure 1 shows the temperature profile of the faulty turbine and the average of the wind farm. The damaged main bearing is evidently warmer than the average. In Figure 2, other evidences of the failure are visible, the relation between main bearing temperature and wind speed is steeper for the defective turbine. Moreover, the density distribution of the faulty main bearing is shifted to higher values than the wind-farm average.

The paper is organized as follows. Section 2 is a review of previous works available in the literature. Section 3 provides an explanation of the data used and the applied pre-processing techniques. Section 4 details how the solution is built, showing the base components and how they are combined into a single health status indicator. In Section 5, results are presented and analyzed, followed by Section 6, where a discussion is provided. Finally, Section 7 contains the final remarks and recommendations for future work directions.

## 2. Previous Works

Various solutions are available to assess the status of wind turbine components and predict failures. Methods can be classified by the type of data utilized. Vibrations, currents and acoustics measurements are particularly effective to diagnose drive-train failures, as documented in [14,15,16,17,18]. These solutions require the installation of additional sensors or in-situ measurements campaigns to collect the data. On the contrary, SCADA system is available as standard equipment for most turbines and its recordings registered in the databases of wind farm owners, such that operators who did not think in advance of data-based predictive maintenance strategy can implement one, using SCADA data, at minimal additional costs.

SCADA predictive maintenance algorithms can be sorted in multiple categories as proposed by Tautz and Watson [6]. In this paper, the following are analyzed:Signal Trending;Normality models;Anomaly detection and Clustering methods.

These three methods are reviewed in the following Section 2.1, Section 2.2 and Section 2.3. Then, a review of Ensemble Learning is provided in Section 2.4 since this is an essential component of the methodology. Relevant applications in predictive modeling and data analysis are discussed. Gradient Boosting and Isolation Forest are also presented in Section 2.5 and Section 2.6, respectively, as they are used in our solution.

### 2.1. Signal Trending

The signal trending approach is based on the study of changes and trends in a long period of time. The underlying hypothesis of this approach is that failures have a sort of signature that can be detected observing variables such as temperatures.

Astolfi et al. proposed a simple, but effective methodology to monitor turbine components. The relation between binned active power and key sensor’s readings such as rotor and generator bearing temperature are tracked within the wind-farm and through time obtaining useful visualization of the state of the turbines and an effective failure detection tool [19]. Cambron et al. proposed a control chart monitoring algorithm based on the comparison of turbines against wind-farm average to detect problems in the generator [20]. Yang et al. presented a technique to track incipient failures through the analysis of the relation between some key variables and contextual parameters such as the wind speed, as shown in the two case-studies the progression of failures is gradual through time and trends towards anomalous conditions can be observed [21]. Feng et al. devised a failure detection strategy for gearboxes based on the thermodynamics and physical behavior of this component, a relation between the loss of efficiency and increase in temperature is derived and utilized to analyze a known failure [22]. Li and Yu formulated a method based on the difference of the median of each turbine with the rest of the wind-farm and used it to build a condition vector. The authors use monitoring charts to generate alarms and discuss several strategies to deal with autocorrelation of operation data [23].

Main advantages of these methods are: ease of implementation, straightforward interpretation of the results and limited data requirements. Being based on simple statistics they can be replicated with minimal knowledge of advanced algorithms and data-analysis techniques. Moreover, the underlying hypotheses of these methods are rooted in the thermodynamics and physical principles governing operations of the components. Wind-farm maintainers often track the same deviations and trends that are automatized by these algorithms, thus results will sound familiar and understandable.

That being said, many of these methods are univariate and are not capable of capturing the interactions between multiple variables. Being wind turbines complex systems, based on the interconnection of mechanical, electrical and electronic components this limitation can be significant. Moreover, incorporating the influence of external variables, such as wind speed and external temperature is not trivial for these methods.

### 2.2. Normality Models

Normal Behavior Modeling (NBM) is a class of predictive algorithms attempting to infer the relation between a set of inputs and a target variable under normal operation of a turbine component. Deviations between predictions and measurements of the target sensor are used to detect failures.

Schlechtingen and Santos compared simple regression models to more sophisticated implementations based on neural networks; details on the training and utilization of normality models are also provided [24]. Puig et al. presented a normality model for turbine generator and gearbox based on Extreme Learning Machines that can be deployed in the cloud, allowing real-time operations [25]. Zhang and Wang proposed an artificial neural network solution for fault detection in wind turbines main bearings, using SCADA data and able to anticipate failures, allowing to schedule maintenance avoiding unexpected breakdowns [26]. A self-evolving maintenance scheduler, based on artificial neural network tracking gearbox bearings conditions is discussed by Bangalore and Tjerrnberg [27]. Normality models are a well established solution in wind turbines’ predictive maintenance field.

The multivariate nature of this approach is suited to capture complex relations between turbines’ sensors, advanced algorithms and neural network architectures can be used to detect non-linearities in the data and model turbine behavior.

Two main criticisms can be addressed to normality models. First, the interpretability of the predictions is scarce as often sophisticated algorithms are used and the influence of input parameters on the output prediction is not trivial, the behavior is that of a ‘black-box’. Second, the selection of the training set to feed to the algorithm is crucial. This task is time-consuming, the sample of data should include all possible operating and external conditions, thus training set shorter than one year are not particularly reliable. On top of that, normal operating conditions only should be selected, this involves a thorough analysis of the turbines logs and eliminations of alarms and unusual operating instances.

### 2.3. Anomaly Detection and Clustering Methods

Anomalies in SCADA data can be detected modifying NBMs. Instead of predicting the value of a target variable using regressive models, the physical model underlying input variables can be learned and the difference between the original and reconstructed signal tracked. Autoenconders (AE), Restricted Boltzmann Machines (RBM) and Generative Adversarial Networks (GAN) are suited for this task [28,29,30]. Signal reconstruction algorithms are capable of capturing non-linearities and produce refined models of the data. On the other hand, as for NBMs, a training set composed of normal operation data is needed. Moreover, complex structures such as AE and GANs often require large volumes of data.

Clustering offers an alternative approach, data is analyzed in search of meaningful groups that can capture interesting relationships within the input variables. Blanco et al. presented a methodology based on Self-Organizing Maps (SOM) and clustering to assess wind turbines’ health status [31]. Du et al. also proposed a SOM based solution to identify system level anomalies [32]. These methods are able to produce insightful representations of the data, that can help the analyst to discover unexpected, but interesting relationships. The purely unsupervised nature though, leads to significant problems in the integration of these algorithms in automatic predictive pipelines. Rules, thresholds and other solutions are needed to make these solutions valuable in an online system.

A large selection of Machine Learning algorithms can also be used for anomaly detection. McKinnon et al. have studied the performances in condition monitoring of a gearbox of three popular algorithms: Isolation Forest (IF), One Class Support Vector Machine (OCSVM) and Elliptical Envelope (EE) and found that depending on the conditions OCSVM and IF reach best results [33]. Purarjomandlangrudi et al. used Support Vector Machine (SVM) to process previously extracted features of the data for early detection of anomalies [34]. Isolation Forest is a particularly interesting approach as it does not require a normal operation dataset to characterize data, anomalies are determined analyzing the density of data in the different regions of the feature space [35]. On top of that, these methods can deal with multivariate distributions and normally require less data and training time with respect to more complex Deep Learning solutions.

### 2.4. Ensemble Learning

Predictions of base learners, sufficiently independent from each other, can be combined into a meta-predictor which often achieves better performances than the individual predictors. This approach is typically referred to as Ensemble Learning, some of its declinations are: boosting, bagging, model averaging and stacking.

This learning paradigm is particularly popular in data-science competitions, a famous example is the algorithm that won the “Netflix Challenge” [36]. An example of Ensemble Learning in an industry application is presented by Wu et al. that used ensembling to deal with imbalanced datasets [37]. A meta-learner trained on a subset of base predictors has been used to improve wind power production in [38,39]. Liu et al. proposed a solution to detect wind turbine blades icing combining features extracted by Deep-Autoencoders into an ensemble model where decision is taken by majority vote [40]. Ensembles can be used to merge information from different data sources, as Turnbull et al. demonstrated using a OCSVM to combine NBMs of a temperature SCADA and vibration data for gearbox and generator bearings of wind turbines [41].

Most of the aforementioned literature make use of a meta-algorithm trained on the predictions of base learners. To do so, a subset of the data have to be withhold to train the higher level algorithm and adjust its parameters. Work orders are used to label healthy and faulty operating conditions of turbines. In this research, an alternative approach is taken, instead of training a high-order classifier, the predictions of the individual unsupervised algorithms are combined into a single health status indicator, to avoid the necessity of labeling data.

### 2.5. Gradient Boosting

First introduced by Friedman, gradient boosting machine is a popular Ensemble algorithm applied both in classification and regression problems [42]. This technique makes use of base-learners, typically decision trees, to learn the relation between input and output data.

The algorithm is iterative as new base learners are routinely trained on a dataset. The name gradient boosting encapsulates the key idea of this technique: accelerating the convergence towards the optimum set of parameters that minimizes the adopted loss function.

Concretely, at each new iteration residuals between prediction and real values are calculated and larger weights are assigned to the instances where the error is greater such that more efforts will be made to fit the model to them. The process is repeated until a stopping criteria, such as the maximum number of iterations or the minimum error, is reached. The algorithm from the original paper [42] is reported below.

ALGORITHM: Gradient Boosting

Given input data ${\left(x,y\right)}_{i=1}^{N}$, a differentiable loss function L(y,ρ), a base learner h(x,a), a function F(x) to estimate and a maximum number of iterations *M*.

These are the steps to follow:$F_{0}\left(x\right)=\arg\;\min_{\rho }\sum_{i=1}^{N}L\left(y_{i},\rho \right)$For m=1 to *M* do:$\tilde{y_{i}}=-{\left[\frac{\partial L(y_{i},F\left(x_{i}\right))}{\partial F(x_{i})}\right]}_{F\left(x\right)=F_{m-1}\left(x\right)},i=1,N$$a_{m}=\arg\;\min_{a,\beta }\sum_{i=1}^{N}{\left[\tilde{y_{i}}-\beta h\left(x_{i};a\right)\right]}^{2}$$\rho_{m}=\arg\;\min_{\rho }\sum_{i=1}^{N}L\left(y_{i},F_{m-1}\left(x_{i}\right)+\rho h\left(x_{i};a_{m}\right)\right)$$F_{m}\left(x\right)=F_{m-1}\left(x\right)+\rho_{m}h\left(x;a_{m}\right)$endForend Algorithm

The algorithm works with a large selection of loss functions and guarantees short training and predicting times. Variations such as XGBoost and LightGBM exist to address some shortcomings of the original algorithm, granting parallel execution and more tuneable parameters.

### 2.6. Isolation Forest

This algorithm was introduced in 2008 by Liu et al. [35]. The founding principle of this method is that anomalies are usually a minority within the data and can be easily divided from the rest of the dataset. With this in mind, multiple fully developed randomized trees are fully trained, meaning that each of their terminal leaf is to be composed of one point only.Trees splits are made setting a random threshold, instead of the optimal one.

Being an Ensemble method this procedure is repeated multiple times, training an entire forest of decision trees. The average path length, meaning the number of splits necessary to isolate a given point, is used to define an anomaly score defined in Equation (Equation 1):(1)$$s\left(x,\psi \right)=2^{\frac{-E(h(x))}{c(\psi )}}$$
where E(h(x)) is the average value of the path length for a given point, c(ψ) is the average path length of unsuccessful search in Binary Search Trees and ψ number of instances. Values of *s* approaching 1 are related to anomalies, scores lower than 0.5 are associated with normal observations and finally, if the entire dataset has scores close to 0.5 no evident anomalies are present.

## 3. Data

SCADA data (10 min time resolution) of two onshore wind farms are used. More than two years of operation are analyzed for a total of 84 turbines. The first wind farm, located in North America, is made of 66, 1.5 MW rated power turbines; the second one, situated in Poland, has 18, 2 MW turbines. SCADA data comes in comma-separated values (csv) format files. The dataset and pre-processing steps are discussed in the following subsections.

### 3.1. SCADA Dataset

The original SCADA dataset is composed of hundreds of columns, since turbines are typically equipped with a multitude of sensors monitoring various components. These sensors record the state of the system at a high frequency. Then, they are downsampled to lower resolution, most commonly 10 min. Raw signal is summarized by taking its mean, standard deviation, minimum and maximum value during the aggregation period. An example of the SCADA dataset is presented in Table 1. In this research, only the main bearing temperature sensor, active power output, environment temperature, wind speed and rotor speed are used, reducing significantly the dimensionality of the dataset. The choice of these variables is dictated by the necessity to characterize the main-bearing working conditions and the context in which it is operating. The relevance of the variable selection has been certified by experts of the wind turbine maintenance field.

### 3.2. Data Processing

Real-life data is typically affected by missing records or outliers, caused by miss-communications or defects of the sensors. A preliminary filter of absurd readings is necessary to reduce the chances of generating false alarms. In the Literature various data filtering approaches have been proposed, most of them are based on the application of statistical filters [43]. In this research a manual threshold values based on technical knowledge of turbines behaviors are used to filter data, as the number of variables to analyze is limited. Values trespassing the imposed thresholds have been removed from the dataset, no imputation nor interpolation are used to fill the gaps.

## 4. Methodology

The scheme of the proposed solution is illustrated in Figure 3. The three indicators used to analyze the data are the following: Mean average temperature of the main bearing; Normality model; and Anomaly detection algorithm.

Each indicator is calculated from raw data at 10 min resolution, using the rest of the wind farm as meter of comparison, a similar approach is used in [19,23,44]. Turbines belonging to the same wind farm are typically from the same manufacturer and technology. Moreover, with regard to external conditions, measurements registered at each turbine such as wind speed and external temperature behave similarly for a given period of time. Results are aggregated on a weekly basis to account for timely variation of conditions between turbines that could skew results excessively. The decision of the weekly aggregation time-frame is dictated by a compromise between ensuring continuous and precise monitoring of turbines and avoiding to flood maintainers with updates on the wind-farm status. The final assessment of the main bearing status is given by the comparison between the averaged value of the combined indicator over a 4 week period and a decision threshold.

A sliding window, as shown in Figure 4 is used to scan the data. On the left side, the normality models rolling scheme, train and test sets are illustrated. On the right side, the rolling window used for the the other two indicators, whose output is calculated directly on the analyzed data, without the need of a training phase, is shown.

### 4.1. Mean Average Indicator

The first indicator tracks the weekly mean average temperature of turbines’ main bearing. This indicator is used to determine whether some turbines are operating at consistently higher temperatures with respect to the wind farm. As presented in Section 1.1, higher temperatures of the main bearing are a common pattern in faulty turbines. An example of the temperature distribution of main bearings is presented in Figure 5. Variation between the turbines is evident.

This indicator is straightforward and easy to interpret, but being the measure of a univariate series, it cannot account for crossed relations between variables such as different operating conditions of the turbines. Higher temperatures may be caused simply by higher production conditions.

### 4.2. Normality Model

Normality models are used to infer the relation between some inputs and a target variable, that can characterize the system under analysis. Normal operating data is needed to train the algorithm and infer the expected behavior of the system. The trained model can be used to predict values that are compared to the measurements of the target variable. Large deviations between predicted and observed values are to be considered suspicious, as they represent deviations from normal behavior.

The pre-selection of normal data is a time-intensive task as it requires the analysis of the work order logs to remove faulty data and abnormal conditions. Automating this task is not trivial and retrain is needed after repairs and modifications of the component. This research presents an adaptation of normality models that allows to skip the labeling step, reducing greatly time overheads in the training phase of the model.

A rolling window, as the one shown in Figure 4 is slid over data, its size being 8 weeks for the training set and 1 week for the test set. The window is then shifted by intervals of one week for next predictions. Instead of mapping the normal behavior of the turbine, the recent relation between the input and target variables is inferred during the training phase.

Deviations in this case, help to detect drifts in the target variable distribution as this is a pattern observed in main bearing failures. Obviously, difference between prediction and observed records can be the consequence of external conditions (high winds, heat waves, etc.) novel to the train set, in this case though a systematic error is expected in all turbines and alarms are unlikely to be raised, as all turbines will have large deviation.

The inputs used for this algorithm are:Active power [W];Wind speed [m/s];Rotor speed [rpm];External temperature [°C].

The main bearing temperature [°C] is used as output.

The sklearn implementation of gradient-boosting regressor for Python programming language is used [45,46]. The number of trees is set to 100 and their depth limited to 2, all other parameters are left to their default values. These parameters are found running cross-validation trials on a subset of the data. Deviation between a predicted and an observed value is measured calculating the root-mean squared error (RMSE), defined by Equation (Equation 2), where $\hat{y_{i}}$ and $y_{i}$ are the predicted and the measured value, respectively, and *N* is the number of instances analyzed:(2)$$RMSE=\sqrt{\frac{\sum_{i=1}^{N}{\left(\hat{y_{i}}-y_{i}\right)}^{2}}{N}}$$

An example of the predictions for a given week and the RMSE by turbine is presented in Figure 6. Error is not uniformly distributed for the different operating conditions, what is important though is the comparison within the wind farm. Turbines that deviate more are isolated from the rest.

### 4.3. Anomaly Detection

Isolation forest algorithm is used to detect anomalies in the windfarm data. Unlike other indicators that model turbines independently, the whole windfarm is analyzed at once with the objective to determine turbines that are behaving differently from the rest.

The feature space is composed by:Rotor speed [rpm];External temperature [°C];Main bearing temperature [°C].

Sklearn implementation of isolation forest is used [47], the percentage of anomalies is set to 10% of the data. This value is chosen after a series of tests on the sample of data. Choosing a higher percentage of anomalies will result in a larger number of normal points being considered as anomalies. A low value, instead, would lead to the isolation of very anomalous working conditions, missing other that can be relevant. A different dataset might require another value for this parameter, thus test of various values and examination of the indicator results are warmly recommended.

As for the other indicators, anomalies are calculated on a rolling-fashion, following the train-predict shown in Figure 4. Once anomalies are found, the percentage of anomalous records with respect to the total number of records for each turbine is calculated, see Equation (Equation 3), where $AS_{i}$ is the anomaly score of turbine *i*, $\hat{x_{i}}$ is the number of anomalous points found for this turbine and $x_{i}$ is the total number of points of the turbine.
(3)$$AS_{i}=\frac{\hat{x_{i}}}{x_{i}}$$

This value is the anomaly detection indicator shown in Figure 7. Turbines having high percentage of anomalies are behaving differently with respect to the wind farm, thus should be more reasonably suspected to have some sort of problem. The right side of Figure 7 illustrates how isolation forest tends to separate data lying in peripheral regions of the feature space, where density of points is typically lower. On the left side, the percentage of anomalous points in each turbines for a given week is shown.

### 4.4. Indicators Merge Processing

The results of the individual algorithms are merged, obtaining a composed score of the turbine status. For each indicator is created a weekly ranking, assigning the percentile of the wind-farm distribution in which each turbine falls.

The three algorithms are designed to assign higher values to turbines, that according to their definition are to be considered faulty. The composition of the three values is calculated using a rolling average, with a sliding window of size 4 weeks as shown in Figure 8, using Equation (Equation 4). Where $x_{ij}$ is the value of indicator *j* for a given turbine in week *i*.
(4)$$H_{ind}=\frac{1}{N_{week}N_{ind}}\sum_{j=1}^{N_{ind}}\sum_{i=1}^{N_{week}}x_{ij}$$

Once the composed score is found, a decision threshold that decides if maintenance is defined. Setting the threshold is a trade-off between anticipating failures and having to do more maintenance intervention. A cost-benefit analysis is recommended to set this value to the value that maximizes economic savings, due to lack of information of the specific costs it has not been possible to optimize in such a way this parameter. A sensitivity analysis of the results is proposed instead.

## 5. Results

Predictions for roughly two years of data are made and evaluated using the work orders logs. Windfarm operators commonly keep track of the checks and interventions required by the turbines. Unlike SCADA datasets, work orders logs do not follow standard formats. Records are typically organized as free-text. The time of the intervention, as well as the affected turbine and information regarding the actions taken are reported. Often, work order logs are used to filter data, removing abnormal operating conditions and assigning a healthy/faulty status to turbines. This research avoided this step, as the absence of a common standard makes difficult to automatize the labeling process; unsupervised algorithms have been favored instead. Work orders have been used only to assess the veracity of the predictions. The work order logs of the failures occurred during the period of analysis is presented in Table 2.

A limit to the anticipation period is defined, as an alarm is useful in practical terms only if it anticipates failures by a margin of time that allows wind farm operators to organize the replacement of the main bearing, optimizing the logistics and minimizing energy losses due to unexpected stops of the turbine. Weekly predictions are grouped in blocks of 4 months, if one alarm occurred during this period the turbine is reported for a maintenance check.

Performance of the proposed methodology is assessed by a confusion matrix. Predictions are sorted in the following categories:True Positive (*TP*);False Positive (*FP*);False Negative (*FN*);True Negative (*TN*).

A *TP* is assigned whenever an alarm is raised and the work order log reports a problem with the main bearing, if no problem is detected a *FP* is marked instead. In case a failure occurs and no alarm is raised, a *FN* is assigned. Finally, when no failure occurs and no prediction is given a *TN* is assigned.

### 5.1. KPIs Definition

A selection of performance indicators is used to track results, namely: accuracy, precision and F1 score. Their definition is defined using Equations (Equation 5), (Equation 6) and (Equation 7), respectively.
(5)$$Accuracy=\frac{TP+TN}{TP+TN+FP+FN}$$
(6)$$Precision=\frac{TP}{TP+FP}$$
(7)$$F1=\frac{TP}{TP+\frac{1}{2}\left(FN+FP\right)}$$

### 5.2. Decision Threshold Sensitivity Analysis

As mentioned in the methodology, the decision threshold is an important parameter. It has a great influence on the results. A sensitivity analysis is proposed, in which the dependence of KPIs with respect to the decision threshold value is studied. The results of this analysis in the two wind-farms are shown in Figure 9.

Firstly, it should be observed that merging the information of the three indicators generally leads to improved performance, regardless of the decision threshold. Except for low values of the threshold, that have no practical relevance, since they would lead to an excessive number of reviews of the turbines.

Secondly, the algorithms are able to separate faulty turbines from healthy ones such that high decision threshold can be set. A high decision threshold means that only the most critical turbines will need checks and most of these reviews lead to the discovery of relevant problems, rather than false alarms.

That being said, a rigorous evaluation of the benefits and costs of choosing a certain value for the decision threshold is recommended to wind farm operators interested in this predictive algorithm. The cost of false alarms and unnecessary checks should be compared to the savings of early fault detection of a main bearing, and an economic optimum searched.

### 5.3. Comparison of Individual and Composed Indicator

The combination of the predictions of multiple algorithms leads to a better overall performance and this is one of the main claim of this research. This observation has been utilized in multiple fields of research, but not frequently by the wind energy predictive maintenance community. Having observed Figure 9, the decision threshold is assigned a value of 0.95 and a comparison of the available indicators and their composition is presented in Figure 10 and Table 3.

Combining predictions of individual indicators into a composed predictor is beneficial according to all the tracked metrics. Precision and F1 score benefit greatly from the combination of the indicators. For wind farm 1, precision and F1 scores double with respect to each single indicator as an effect of decreased number of FP, combining various sources allows to discard behaviors that are unusual, but not so critical to deserve maintenance check. Wind farm 2 also manifests an increase of precision and F1, but not as large as wind farm 1, overall results are better though as a precision of 33.6% and F1 score of 50.3% are reached. Accuracy is the metric that less benefits from the merging process as the starting values are already high, but an increase of 3–5 percentage points is recorded.

The information fusion process increase complexity of the predictive algorithm, but grants improved performance. Moreover, the design of simpler and specialized algorithms that focus on the detection of specific patterns in the data helps interpretability of the predictions. Base algorithms are implemented with the objective of capturing a specific trend in the data, rather than searching generic relationships within the variables. Once an alarm is raised the analyst can assess which indicators have greater influence in the alarm and verify whether the prediction is reasonable and eventually schedule a check of the turbine.

Information fusion theory and Ensemble learning state that a combined indicator performs best when its basic components have little correlation between themselves, as indicators mutually overcome each others shortcomings. The scatter-plot and correlation matrix of the indicators is presented, respectively in Figure 11 and Figure 12.

The correlation coefficient of the indicators is never greater than 0.4. The amount of overlapped, redundant information is small, thus making their combination beneficial for overall predictive performances. Whenever additional indicators are added their correlation with the existing predictors should be checked. If two indicators are too similar, then only one should be used and the other may be discarded.

### 5.4. Failure Anticipation

Predictions, to be useful, must anticipate a failure by a sufficiently large amount of time, giving to the wind farm operator the possibility to organize the substitution of the broken component and adjust turbines production not to incur in fines due to missed production.

The verification of the anticipation margin is made observing a heatmap representation of the value of the combined indicator for the two analyzed wind farm. The combined indicator for wind farm 1 is shown in Figure 13. Two failures occurred and both of them are preceded by various weeks of high scores of the fault indicator value. A minimum of one month of anticipation of the main bearing failure is ensured.

Figure 14 presents results for wind farm 2. Both failures are correctly predicted with a safe margin of time allowing maintenance to be timely organized. Both heatmaps show turbines with high values of the combined indicator, without recorded maintenance interventions. This can be caused by concurring failures in other components or different operating conditions with respect to the rest of the wind farm. That being said, the ratio between false positives and true positives indicates that the proposed methodology offer a valid solution to automatize turbine reviews.

## 6. Discussion

The proposed solution is characterized by an increased complexity of the decision process, when compared to Signal Trending or Normal Behavior Modeling techniques, as the information of multiple indicators is considered. Choosing a complete and significant set of indicators might be challenging, that being said, the presented results prove that it is a beneficial choice.

This strategy is highly modular, new indicators tailored to capture different behaviors of the data or utilizing other data streams can be easily incorporated into the decision process, once their complementarity to the already included indicators is verified. The use of multiple indicators based on detection of specific patterns in the data provides a more explainable interpretation of the behavior of the turbine with respect to complicated solutions processing data in a unique algorithm, often based on black-box structures.

The chosen indicators worked especially well for the detection of main bearing failure. As presented in Section 5.2, it is possible to set a high value of decision threshold without undermining failure detection. This means that wind farm operator do not require to check too many turbines to be sure to anticipate failures, a small number of revisions are necessary, following this strategy, and most of them will result in the discover of defects.

While more complex, the use of various indicators, proved especially beneficial in terms of elimination of FPs, as clearly shown in Section 5.3. Precision and F1 score greatly take advantage of the use of multiple indicators. In the first wind farm precision and F1 score almost doubled their values. The second wind farm benefits in a lower measure of the merging process, but significant improvements are observed.

Another remarkable characteristic of this approach is the ability to reliably anticipate failures, as debated in Section 5.4. It is critical to guarantee a margin of anticipation for main bearing failures, as the logistic is not trivial and a maintenance intervention cannot be arranged on a short-notice. As it is shown, the predictive methodology anticipated all four events by at least one month. Wind farm operators are then put in condition to adapt their production schedule and avoid losses due to unexpected and critical failures of main bearings.

Ultimately, the decision to avoid supervised learning solutions that require the time-consuming phase of data labeling helped to decrease greatly setup times of this architecture, repaying the additional time required to implement a set of multiple indicators and a merging strategy to aggregate their results.

## 7. Conclusions

This paper proposes a novel and innovative predictive maintenance solution based on Ensemble Learning using SCADA data, for wind turbine farms. The main characteristics of this solution can be summarized in three key-points:Unsupervised algorithms;Interpretable results;Combination of various indicators into a more reliable one via Ensemble Learning.

The time to pre-process and train algorithms is greatly reduced, as labeling of operating data into healthy and faulty conditions is not required. Incidentally, this techniques also has more flexible requirements, work orders are not necessary as they are used for evaluation purposes only. The presented algorithm only requires SCADA data to be put into production.

The indicators are designed on specific failure patterns, that are easy to interpret (drift in temperatures, changes in the relation of key variables...). The presented methodology has been rigorously tested on two year worthy of data from two onshore wind farms, for a total of 84 turbines.

Results proved that the combination of multiple indicators into a single predictor grants substantial improvements in performances, reaching an average accuracy of 95.1%, precision of 24.5% and F1 score of 38.5%. The sensitivity to key parameters as the threshold that discriminate faulty turbines from normal ones is studied, suggesting that high threshold values leads to good results, as the chosen indicators are able to isolate faulty from healthy turbines. The anticipation of failure, in all four events analyzed, is no less than one month giving wind farm operators time to organize logistics and minimize losses related to downtime.

Future researches may design additional indicators, as well as define tuning strategies of the decision threshold, incorporating maintenance costs and savings for early fault detection and optimize economic benefit of the predictive strategy. If vibration or acoustics data is available, new indicators could be designed and integrated to improve performances. It has to be noticed that we have been able to test this methodology on main bearing failures only, due to the limitations of the dataset at hand. Other turbine systems, such as gearbox and generator bearings or pitch actuators could have different failure signatures, thus other indicators might be needed and adjustments to the presented methodology required. The application of this strategy to monitoring of other components is a line of research that we warmly recommend to readers.

## Figures and Tables

**Figure 1 sensors-21-01512-f001:**
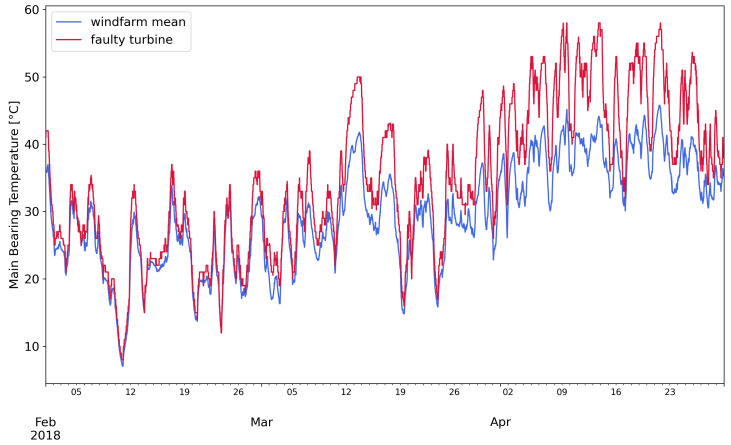
Timeseries profiles of the main bearing temperature of a faulty turbine and the average of the wind-farm.

**Figure 2 sensors-21-01512-f002:**
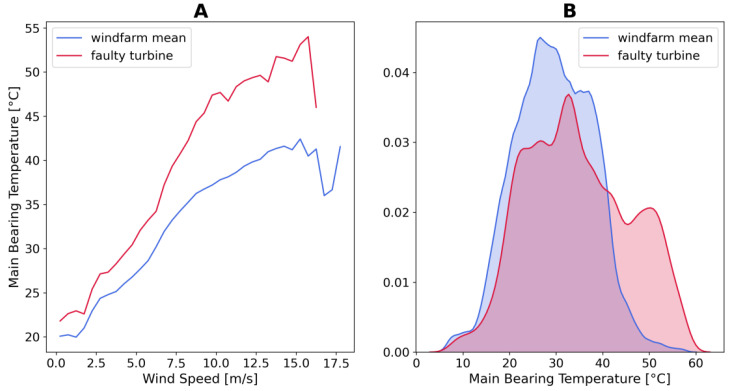
(**A**) Relation between main bearing temperature and wind speed. (**B**) Probability density plot of the main bearing temperature of a faulty turbine and the average of the wind-farm.

**Figure 3 sensors-21-01512-f003:**
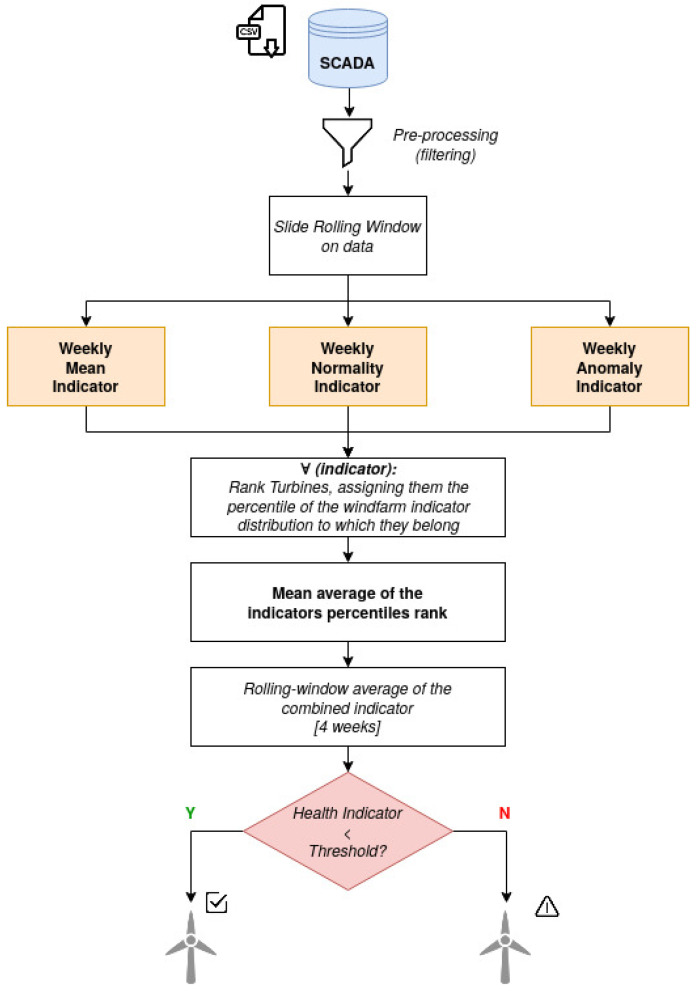
Diagram of the predictive maintenance solution.

**Figure 4 sensors-21-01512-f004:**
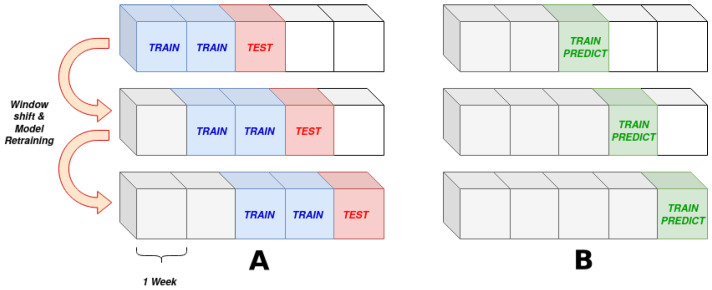
(**A**) The rolling window train/test scheme used for normality models. (**B**) The rolling window train/test scheme used for mean and anomaly indicators.

**Figure 5 sensors-21-01512-f005:**
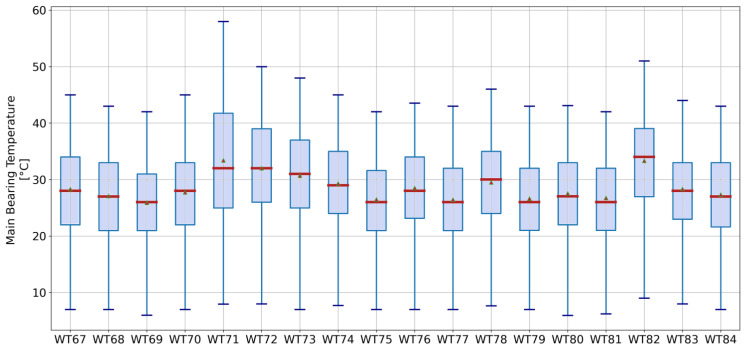
Boxplot of the main bearing temperature. The median is represented by the red line and the mean corresponds to the triangle.

**Figure 6 sensors-21-01512-f006:**
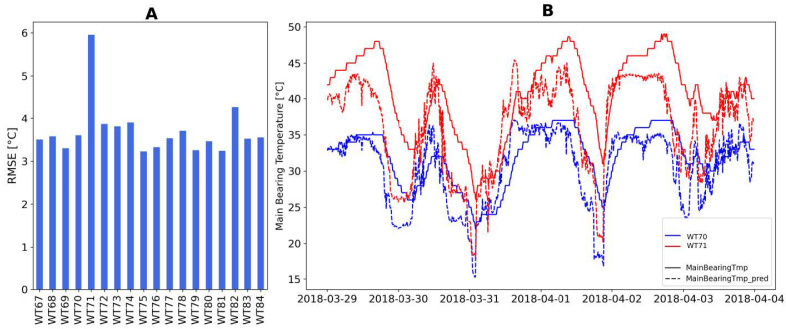
(**A**) Normality indicator, RMSE by turbine. (**B**) Timeseries comparison of predicted versus measured value.

**Figure 7 sensors-21-01512-f007:**
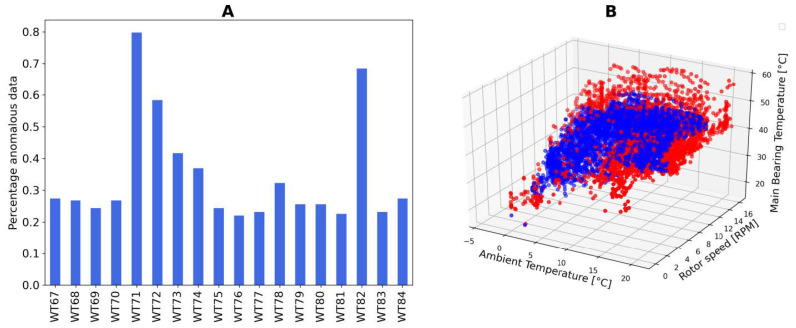
(**A**) Anomaly Indicator plots: percentage of anomalous versus total number of points. (**B**) 3D plot showing normal (blue) versus anomalous (red) points.

**Figure 8 sensors-21-01512-f008:**
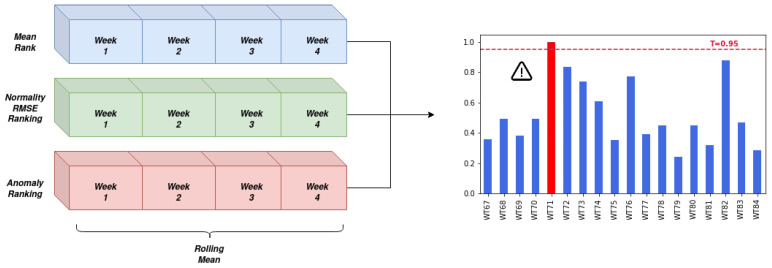
Composed indicator calculation scheme and decision threshold setting.

**Figure 9 sensors-21-01512-f009:**
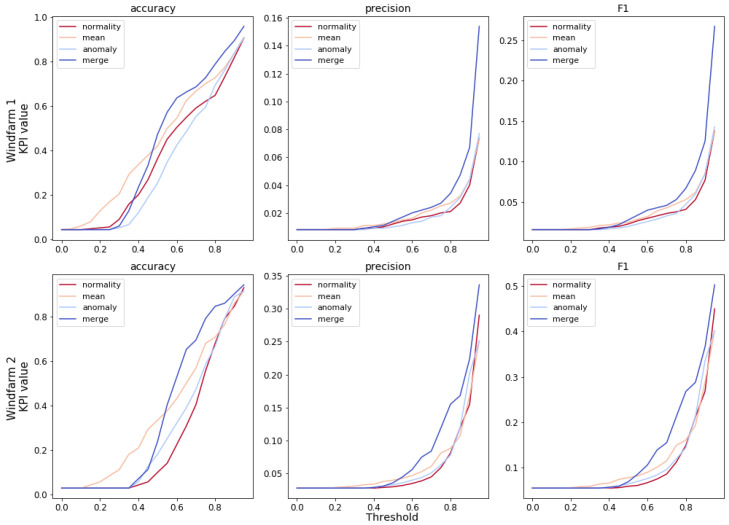
Relation between KPIs and decision threshold value by wind-farm and indicator.

**Figure 10 sensors-21-01512-f010:**
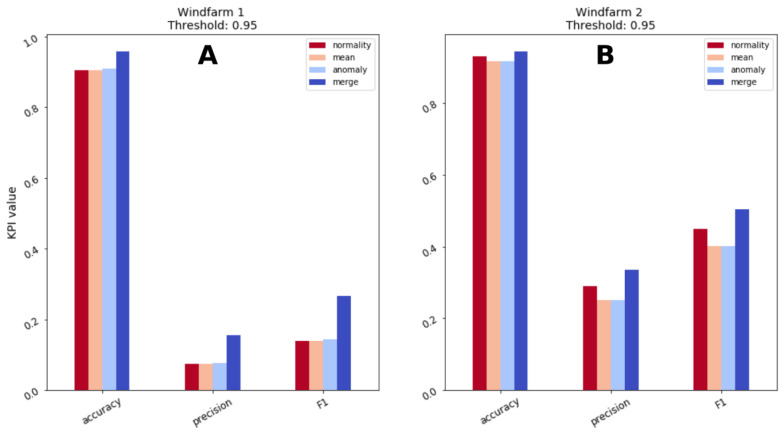
(**A**) Performance comparison of individual and composed indicators for Windfarm 1 and (**B**) Windfarm 2.

**Figure 11 sensors-21-01512-f011:**
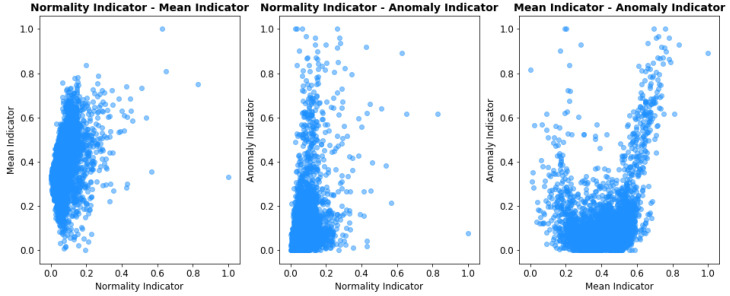
Scatter-plot of each pair combination of basic indicator.

**Figure 12 sensors-21-01512-f012:**
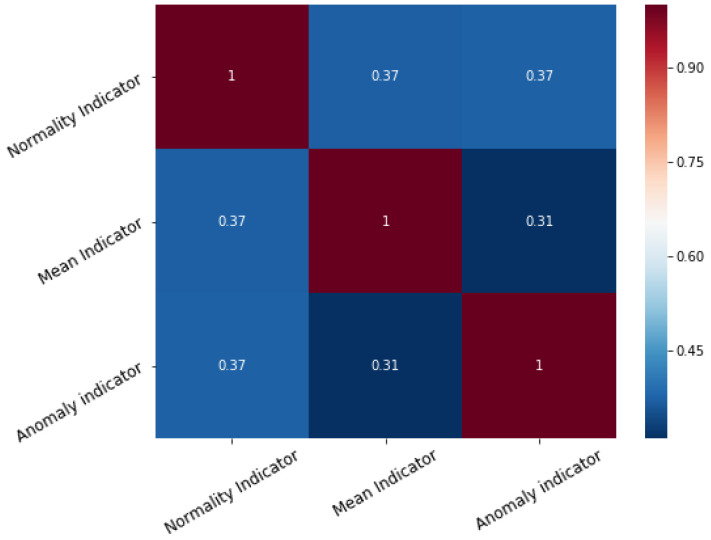
Correlation matrix of the base indicators.

**Figure 13 sensors-21-01512-f013:**
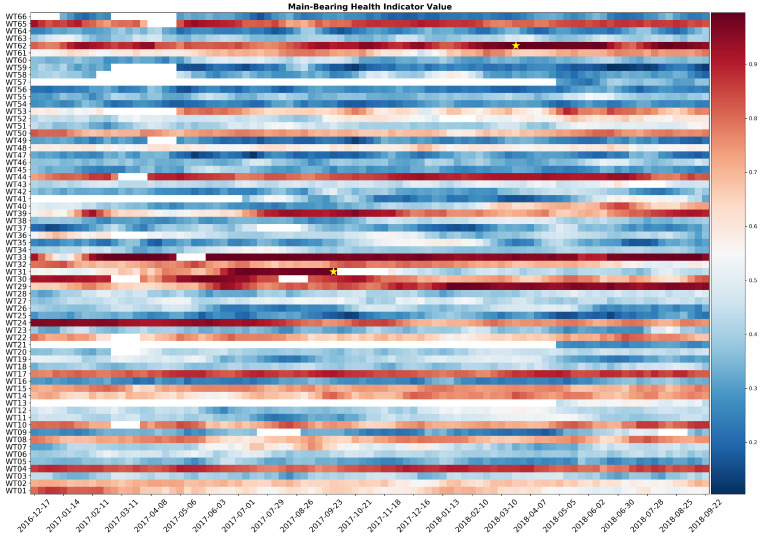
Heatmap of the combined main bearing health status indicator for wind farm 1. Failures are represented by a yellow star.

**Figure 14 sensors-21-01512-f014:**
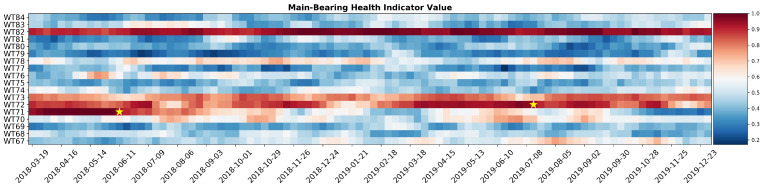
Heatmap of the combined main bearing health status indicator for wind farm 2. Failures are repersented by a yellow star.

**Table 1 sensors-21-01512-t001:** Sample of SCADA data.

Turbine	Timestamp	Main-Bearing Temp. C${}^{\circ }$	Active Power W	External Temp C${}^{\circ }$	Wind Speed m/s	Rotor Speed rpm
WT01	02/01/18 10.00 am	32	1529	−6	14	17
WT01	02/01/18 10.10 am	32	1532	−6	13	17
WT01	02/01/18 10.20 am	32	1532	−6	13	17

**Table 2 sensors-21-01512-t002:** Main Bearing failures work order logs.

Wind Farm	Location	Failure Date	Turbine	Comment
1	US	7 October 2017	WT31	Main Bearing Replacement
1	US	24 March 2018	WT62	Main Bearing Replacement
2	Poland	11 June 2018	WT71	Main Bearing Exchange
2	Poland	15 July 2019	WT72	Main Bearing Exchange

**Table 3 sensors-21-01512-t003:** Comparison of the results of individual and combined indicator for a threshold value of 0.95.

Windturbine	Indicator	TP	FP	FN	TN	Accuracy	Precision	F1
WF1	normality	48	600	0	5688	0.905	0.074	0.138
	mean	48	600	0	5688	0.905	0.074	0.138
	anomaly	48	576	0	5712	0.909	0.077	0.143
	merge	48	264	0	6024	0.958	0.154	0.267
WF2	normality	49	120	0	1577	0.931	0.29	0.45
	mean	49	146	0	1551	0.916	0.251	0.402
	anomaly	49	146	0	1551	0.916	0.251	0.402
	merge	49	97	0	1600	0.944	0.336	0.503

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
