# Peer review of "An Ensemble Learning Solution for Predictive Maintenance of Wind Turbines Main Bearing"

_sensors, 2021, doi:10.3390/s21041512_

Round 1

Reviewer 1 Report

The manuscript entitled “AN ENSEMBLE LEARNING SOLUTION FOR PREDICTIVE MAINTENANCE OF WIND TURBINES’ MAIN BEARING” deals with a very interesting topic in wind energy literature, which is the use of SCADA data for the diagnosis of main bearing damages.

The manuscript is well written and organized and two test case discussions are included.

I do not have particular remarks about the present paper.

There is just one aspect I would like to indicate to the attention of the authors. A methodology based on the analysis at the level of wind farm (and consequent comparison of each wind turbine against the wind farm trend) has been proposed here: Astolfi, D., Castellani, F., & Terzi, L. (2014). Fault prevention and diagnosis through SCADA temperature data analysis of an onshore wind farm. Diagnostyka, 15. I would suggest to include this manuscript in the references because there are relevant contact points with the proposed study. As you can see, in that manuscript the method is quite simple and consists substantially in the binning method for the power – temperature curves, which are analyzed on multiple time scales for all the wind turbines in a wind farm. In that study, main bearing damages are easily individuated with much advance. Here comes my point: given my experience in SCADA data analysis, I recognize that main bearing damages are the easiest to individuate through SCADA temperature data analysis and, given the complexity and the structure of your method, I would be interested in seeing it applied to more complicated fault diagnosis, as for example regarding generator bearings. Then, my question is if you have other test cases at hand to validate your method: if not, I would recommend to add at least a discussion about this point in the manuscript.

Furthermore, I would like to indicate that the literature review is quite comprehensive but in my opinion it lacks a devoted discussion about studies dealing with SCADA temperature data analysis for fault diagnosis.

Author Response

Responses to Reviewer #1

We, the Authors, would like to thank Reviewer #1 for taking the time to read and criticize

our work. The suggested paper was new to us, but it has revealed very relevant to further

support our approach. It was also an interesting reading and possible material for future

improvements of our predictive algorithms. In the following we have reported the remarks

of Reviewer #1 and our response to them.

➢ Remark #1

“There is just one aspect I would like to indicate to the attention of the authors. A methodology

based on the analysis at the level of wind farm (and consequent comparison of each wind turbine

against the wind farm trend) has been proposed here: Astolfi, D., Castellani, F., & Terzi, L. (2014).

Fault prevention and diagnosis through SCADA temperature data analysis of an onshore wind

farm. Diagnostyka, 15. I would suggest to include this manuscript in the references because there

are relevant contact points with the proposed study. As you can see, in that manuscript the method

is quite simple and consists substantially in the binning method for the power – temperature curves,

which are analyzed on multiple time scales for all the wind turbines in a wind farm. In that study,

main bearing damages are easily individuated with much advance. Here comes my point: given my

experience in SCADA data analysis, I recognize that main bearing damages are the easiest to

individuate through SCADA temperature data analysis and, given the complexity and the structure

of your method, I would be interested in seeing it applied to more complicated fault diagnosis, as

for example regarding generator bearings. Then, my question is if you have other test cases at hand

to validate your method: if not, I would recommend to add at least a discussion about this point in

the manuscript.”

Answer:

The recommended paper proved useful, as Reviewer #1 pointed out, it shares various

similarities with our approach and it is an example of a simple but effective monitoring technique. It

has been included in the “Signal Trending” sub-section of the Literature Review, as well as

additional example of previous works that have used the wind-farm to provide context and

determine anomalies within turbines.

Regarding the second part of the comment, main bearing is indeed a “simple” fault to detect,

due to the typical temperature “signature” that we, as many other Authors, have observed before

failures. We are working on testing this technique on different failures, such as gearbox and

generator bearings. That being said, we do not have sufficient failures in the dataset at hand to

propose a proper assessment of our methodology for other components. As suggested, we have

added a remark in the conclusion, recommending the evaluation of this strategy in other

components and we hope to be able to do it in the future.

The following modifications have been done to the original Manuscript in response to this

comment

[lines 129-133]:

“Astolfi & Al. proposed a simple, but effective methodology to monitor turbine components. The

relation between binned active power and key sensor’s readings such as rotor and generator

bearing temperature are tracked within the wind-farm and through time obtaining useful

visualization of the state of the turbines and an effective failure detection tool [19]. ”

lines[561-566]:

“It has to be noticed that we have been able to test this methodology on main bearing failures only,

due to the limitations of the dataset at hand. Other turbine systems, such as gearbox and generator

bearings or pitch actuators could have different failure signatures, thus other indicators might be

needed and adjustments to the presented methodology required. The application of this strategy to

monitoring of other components is a line of research that we warmly recommend to readers. ”

➢ Remark #2

“Furthermore, I would like to indicate that the literature review is quite comprehensive but in my

opinion it lacks a devoted discussion about studies dealing with SCADA temperature data analysis

for fault diagnosis.”

Answer: We have decided to include 5 additional references addressing the topic of fault monitoring

using SCADA data, with a dedicated focus on temperature signal. Instead of a new Section we

have decided to extend Section 1.1 since there is mentioned the opportunity of using temperature

data to predict failures.

The modifications to the original Manuscript in response to this comment are here reported:

lines [79-91]

“Different authors have successfully studied temperature behaviors to predict failures in various

turbines’ components. Guo & Al. devised a monitoring strategy for turbines’ generators based on

tracking of the generator temperature via change detection of a memory matrix of the component

behavior [9]. Qiu & Al. presented a thermophysics approach to assess drive train conditions from

which various diagnostic rules are defined [10]. Tonks & Wang showed experimentally that

monitoring temperature can reveal misalignments and problems of shaft couplings, as these defects

increase friction therefore temperature of the component [11]. Cambron & Al. developed a method

to monitor main bearing condition comparing the measured and expected temperture of the

component, predictions were obtained using a physical model of the bearing [12]. Sun & Al.

describe an anomaly identification method using mainly temperature readings and other standard

SCADA signals to monitor the behavior of the major components [13].”

Reviewer 2 Report

Comments on the manuscript "An ensemble learning solution for predictive maintenance of wind turbines’ main bearing" by Beretta  et al. submitted to Sensors

In this work, the authors developed an ensemble learning model for predicting the maintenance of wind turbines' main bearing using temperature measurement of SCADA data. The work is well presented. The literature review is comprehensive. Some comments are as follows: 

1. Page 8: "Moreover, external conditions (wind speed and temperature) are not expected to vary greatly". As we all know, these external conditions vary over months, seasons and years. The authors need to justify this assumption and demonstrate how the obtained model and model prediction affected by these external conditions. 

2. Page 11: "the percentage of anomalies is set to 10% of the data.". How is this percentage determined? 

3. The font sizes of axis labels of some figures are small. 

Author Response

We, the Authors, would like to thank Reviewer #2 for taking the time to read and criticize

our work. The three remarks that we have received helped us to make methodology

clearer, specifying important details that in the first version of the Manuscript were not so

straightforward as we deemed. Below can be found the suggestions we have received and

the actions we have taken to address them.

➢ Remark #1

“Page 8: "Moreover, external conditions (wind speed and temperature) are not expected to vary

greatly". As we all know, these external conditions vary over months, seasons and years. The

authors need to justify this assumption and demonstrate how the obtained model and model

prediction affected by these external conditions.”

Answer:

We have better specified what we meant to say with that sentence. The observation that

external conditions do not vary greatly is to be referred to the whole wind-farm for a given moment

in time, not throughout the whole year. As Reviewer #2 correctly pointed out, external conditions

vary greatly throughout the year. This is not a problem though, as our methodology is based on the

comparison of turbines with the rest of the wind-farm at the instant in time. The reformulation of

the mentioned sentence should clarify doubts and prevent misunderstandings.

The following modifications have been done to the original Manuscript in response to this comment

lines [319-321]

“Moreover, with regard to external conditions, measurements registered at each turbine such as

wind speed and external temperature behave similarly for a given period of time.”

➢ Remark #2

“Page 11: "the percentage of anomalies is set to 10% of the data.". How is this percentage

determined?”

Answer: This is a hyperparameter of our solution, is something that we recommend the user to

experiment with, as a different value might be better for a new set of data. Concretely, we have

taken a set of data and tested various values. Our objective was to classify as anomalies a

sufficiently large pool of points, such that alarms are not raised for isolated extemporary conditions,

10% proved a good compromise between having sufficient point per each turbine and not including

too many “normal” points as anomalies.

We would like to remark the fact that even some operating conditions, that could be

considered normal are marked as anomalies is not a major problem. What we are interested to are

differences in the behavior of the turbines, if a turbine has a significantly higher percentage of

anomalous points with respect to the others it is to be reported.

The following modifications have been done to the original Manuscript in response to this comment

lines [382-387]

“This value is chosen after a series of tests on the sample of data. Choosing a higher percentage of

anomalies will result in a larger number of normal points being considered as anomalies. A low

value, instead, would lead to the isolation of very anomalous working conditions, missing other that

can be relevant. A different dataset might require another value for this parameter, thus test of

various values and examination of the indicator results are warmly recommended.”

➢ Remark #3

“The font sizes of axis labels of some figures are small.”

Anwser: We have increased the size of figures labels to make them easier to read.

Reviewer 3 Report

The authors propose an ensemble learning solution for predictive maintenance of wind turbain.

The proposed approach is interesting but there are some points that the authors should better discuss.

The authors should be better described the novelties of their approach with respect to existing ones. In particular, the author should discuss limitation and cons of the examined approaches that they aim to overcome. Furthermore, the authors should provide more details and discussion about the obtained results. The Discussion section also needs to be improved by analyzing the outcome of evaluation section.

I suggest to further analyze more recent approaches about the examined topics. In particular, I suggest the following papers to further investigate big data and deep learning techniques for predictive maintenance in the introduction section:

1)  Deep Learning for HDD health assessment: an application based on LSTM" in IEEE Transactions on Computers, vol. , no. 01, pp. 1-1, 5555.

Finally, I suggest to perform a linguistic revision.

Author Response

Responses to Reviewer #3

We, the Authors, would like to thank Reviewer #3 for taking the time to read and criticize

our work. In response to the comments we have added a Discussion section where

pro/cons of our solution are analyzed. We believe that this addition improved the quality

and clarity of the Research and will prove useful for future readers. Below we have

reported the received remarks and our answers to each of them.

➢ Remark #1

“The authors should be better described the novelties of their approach with respect to existing

ones. In particular, the author should discuss limitation and cons of the examined approaches that

they aim to overcome. Furthermore, the authors should provide more details and discussion about

the obtained results. The Discussion section also needs to be improved by analyzing the outcome of

evaluation section.”

A:

The novelties of our approach are stated in the Introduction [lines 54 – 60] and can be

expressed in three main points:

1. Present an unsupervised system, requiring minimum setup and limited prerequisites, capable to monitor entire

wind farms.

2. Provide interpretable and understandable predictions, in contrast to black-box solutions.

3. Implement an Ensemble Learning strategy that produces reliable predictions from a set of understandable

indicators, improving their individual performances.

As discussed throughout the introduction and Literature Review, no other works were found

presenting a unique and comprehensive methodology, allowing to monitor entire wind-farm fleets

without the necessity of time-consuming pre-processing techniques and providing interpretable

results. Moreover, ensemble learning techniques while very popular in different domain of

applications rarely have been used in wind energy field. Our research showed that these techniques

are effective and are a promising line of research for wind energy predictive maintenance industry.

In the Literature Review presented in Section 2 we have discussed various approaches for

turbine monitoring. Vibrations, current analysis and others at line 107-109. Of these we have

mentioned the fact that are not recorded by standard sensors. They are not often available to windfarm

owners who would require substantial additional investments to apply this monitoring

techniques. Approaches based on SCADA data are thoroughly analyzed in Section 2.1-2.2-2.3.

Previous researches are presented and discussed. At the end of each subsection are available

commentaries of the pros & cons of each solution. [lines 146-157 / lines 199-204 / lines / lines 214-

216].

In Section 5 we have first introduced the data that we have used to evaluate predictions, then

an explication of the tracked metrics is provided in Section 5.1, then a discussion of aforementioned

results is provided analyzing the dependence on decision threshold (Section 5.2), comparison of the

results of the individual indicators and our proposed solution based on their merge (Section 5.3).

Finally, (Section 5.4) provides a presentation of the capability of our solution to anticipate failures.

Following Reviewer #3 comment we have decided to add a Discussion section summarizing

the characteristics, pros and cons of our solution as well as an analysis of the most remarkable

results that we have obtained.

The following modifications have been done to the original Manuscript in response to this comment

lines [506-535]

“The proposed solution is characterized by an increased complexity of the decision process, when

compared to Signal Trending or Normal Behavior Modeling techniques, as the information of

multiple indicators is considered. Choosing a complete and significant set of indicators might be

challenging, that being said, the presented results prove that it is a beneficial choice. This strategy

is highly modular, new indicators tailored to capture different behaviors of the data or utilizing

other data streams can be easily incorporated into the decision process, once their complementarity

to the already included indicators is verified. The use of multiple indicators based on detection of

specific patterns in the data provides a more explainable interpretation of the behavior of the

turbine with respect to complicated solutions processing data in a unique algorithm, often based on

black-box structures. The chosen indicators worked especially well for the detection of main

bearing failure. As presented in Subsection 5.2, it is possible to set a high value of decision

threshold without undermining failure detection. This means that wind farm operator do not require

to check too many turbines to be sure to anticipate failures, a small number of revisions are

necessary, following this strategy, and most of them will result in the discover of defects. While

more complex, the use of various indicators, proved especially beneficial in terms of elimination of

FPs, as clearly shown in Subsection 5.3. Precision and F1 score greatly take advantage of the use

of multiple indicators. In the first wind farm precision and F1 score almost doubled their values.

The second wind farm benefits in a lower measure of the merging process, but significant

improvements are observed. Another remarkable characteristic of this approach is the ability to

reliably anticipate failures, as debated in Subsection 5.4. It is critical to guarantee a margin of

anticipation for main bearing failures, as the logistic is not trivial and a maintenance intervention

can not be arranged on a short-notice. As it is shown, the predictive methodology anticipated all

four events by at least one month. Wind farm operators are then put in condition to adapt their

production schedule and avoid losses due to unexpected and critical failures of main bearings.

Ultimately, the decision to avoid supervised learning solutions that require the time-consuming

phase of data labeling helped to decrease greatly setup times of this architecture, repaying the

additional time required to implement a set of multiple indicators and a merging strategy to

aggregate their results.”

➢ Remark #2

“I suggest to further analyze more recent approaches about the examined topics. In particular, I

suggest the following papers to further investigate big data and deep learning techniques for

predictive maintenance in the introduction section:

1) Deep Learning for HDD health assessment: an application based on LSTM" in IEEE

Transactions on Computers, vol. , no. 01, pp. 1-1, 5555.”

A:

We thank Reviewer #3 for the suggested paper, as it proved interesting and well written.

That being said, it appears out-of-context with respect to this Manuscipt. As it can be seen all the

references that were included refers to application regarding wind-energy industry with the

exception of references to the utilized algorithms [35], [42], [46], [47] and the “Netflix Challenge

winning algorithm” probably the most famous application of Ensemble Learning [36]. It is our

intention to provide Readers with a useful and relevant Bibliography of the wind-energy sector as

this is a quite sector-focused research.

We have also reviewed the included references and it results that most of the references (20

out of 47) reported in the Manuscript were published not earlier than 2018, some have been

published during the last year 2020-2021 (9 out of 47).

Deep Learning approaches are reported:

• [24] presents Artificial Neural Networks including lagged inputs

• [25] presents Extreme Learning Machine applied to wind turbine maintenance

• [16-27] present solution based on Artificial Neural Networks

• [28-29-30] presents respectively an Autoencoder, a Restricted Boltzmann Machine and a

Generative Adversarial Network applied to various turbine failures.

• [31-32] discuss application of Self Organizing Maps for turbine monitoring

• [40] uses Deep-Autoencoders to detect icing on turbines’ blades.

We believe that the addressed references provide a complete and updated overview of

monitoring techniques available in the wind-energy sectors. Simple and more complex techniques

are reported. This selection provides the curious reader a comprehensive map to explore different

approaches to tackle the predictive maintenance problem.

Round 2

Reviewer 1 Report

The authors have addressed all the comments. The manuscript in my opinion is adequate for publication.

Reviewer 3 Report

I think that the authors have addressed all my concerns